# COLD Decoding: Energy-based Constrained Text Generation with Langevin Dynamics

**Lianhui Qin**[1]    **Sean Welleck**[1 2]    **Daniel Khashabi**[3*]    **Yejin Choi**[1 2]
[1]Paul G. Allen School of Computer Science & Engineering, University of Washington
[2]Allen Institute for Artificial Intelligence
[3]Department of Computer Science, Johns Hopkins University

## Abstract

Many applications of text generation require incorporating different constraints to control the semantics or style of generated text. These constraints can be hard (e.g., ensuring certain keywords are included in the output) and soft (e.g., contextualizing the output with the left- or right-hand context). In this paper, we present *Energy-based Constrained Decoding with Langevin Dynamics* (COLD), a decoding framework which unifies constrained generation as specifying constraints through an energy function, then performing efficient differentiable reasoning over the constraints through gradient-based sampling. COLD decoding is a flexible framework that can be applied directly to off-the-shelf left-to-right language models *without* the need for any task-specific fine-tuning, as demonstrated through three challenging text generation applications: lexically-constrained generation, abductive reasoning, and counterfactual reasoning. Our experiments on these constrained generation tasks point to the effectiveness of our approach, both in terms of automatic and human evaluation.[1]

## 1 Introduction

Many text generation applications require producing text that is not only fluent, but also satisfies various constraints which control the semantics or style of the generated text. For example (Figure 1), for knowledge-grounded or keyword-guided generation, we might want to ensure that certain keywords are included in the generated output as *hard* lexical constraints [29, 52]. For other types of text generation, we often wish to incorporate *soft* topical constraints to contextualize the desired output, e.g., abductively [43] reasoning about what happened in the middle of a story given the past and the future story context [1]. Yet another class of text generation applications requires revising an input based on a new counterfactual condition [13], which simultaneously requires semantic coherence as well as *minimal-edit* constraints with respect to the input text [44].

The dominant paradigm to various text generation applications has been supervised learning with task-specific training data. However, different applications require varied and potentially evolving constraints, and annotating a large amount of task-specific training data for each different combination of constraints can be costly. Recent work has explored incorporating constraints through energy-based text modeling that alleviates the need of supervised data [23, 7, 41]. Yet those approaches still require expensive training of specific generation models. In addition, training might not even be feasible with recent models that are extreme in scale, like GPT-3 [3]. This motivates the need to enrich *decoding* algorithms that can work directly with pretrained language models without task-specific fine-tuning, and support complex combinations of hard and soft constraints to control the generated text on the fly.

---

* Work done while working at Allen Institute for AI.

[1]Code is available at https://github.com/qkaren/COLD_decoding

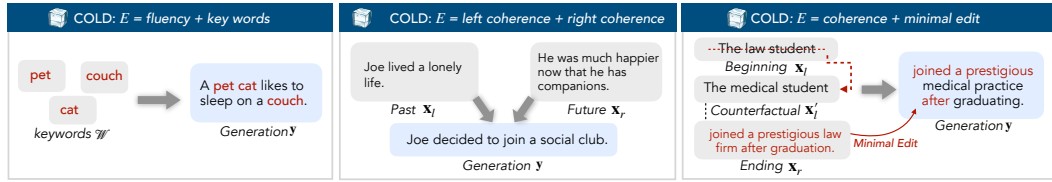

Lexically Constrained Generation      Abductive Reasoning      Counterfactual Reasoning

Figure 1: Applying COLD to different constrained generation tasks amounts to specifying an energy function $E$ by plugging in relevant constraint functions. Text in grey boxes is the input, and text in blue boxes is the output.

We propose a new constrained decoding approach that formulates decoding as sampling from an energy-based model (EBM) [16, 27]. Constrained generation with our approach amounts to specifying an energy function by plugging in arbitrary constraint functions that are suitable for the task at hand, then sampling from its induced distribution. In particular, to overcome the longstanding challenges of sampling discrete text from EBMs, we for the first time introduce Langevin dynamics [53] to text-based EBMs for efficient *gradient*-based sampling. As a result, our approach, *Constrained Decoding with Langevin Dynamics* (COLD), performs sampling by iteratively updating a continuous relaxation of text using gradients of the energy function. The resulting continuous text samples are then mapped back to the discrete space with a simple guided discretization approach, yielding text sequences that are fluent and adhere to the constraints.

Our work makes unique contributions to a recent line of research investigating decoding algorithms for incorporating different constraints [45, 6, 33, 26] in three distinct aspects. First, our formulation unifies various constrained generation scenarios that involve hard lexical constraints and/or soft contextual constraints: specifying an energy function, then sampling from its induced distribution. Second, we propose a *sampling* method, which complements decoding algorithms that look for a single optimal solution. Finally, we provide new empirical insights into the strengths and weaknesses of existing approaches to discrete search and differentiable reasoning.

To test the flexibility and empirical performance of COLD decoding, we experiment with three challenging text generation tasks: lexically constrained generation [29, 18], abductive reasoning [1], and counterfactual story generation [44]. COLD achieves better lexical coverage than NEUROLOGIC [33], a beam-based discrete decoding algorithm specifically designed for lexically constrained generation, while producing more coherent and higher quality text than DELOREAN [45], a state-of-the-art gradient-based generation method for abductive reasoning and counterfactual reasoning. COLD supports all three constrained generation settings under a unified framework – specifying an energy function using a collection of fluency and task-specific constraints, then sampling from its induced distribution and achieves strong performance on both automatic and human evaluation.

## 2   Background

**Neural text generation.** Neural text generation typically involves two stages: modeling a distribution over text sequences, and using a *decoding algorithm* to generate sequences with the model. Let $\mathbf{y} = (y_1, \ldots, y_T)$ denote a discrete sequence where each $y_t$ is a token from a vocabulary $\mathcal{V}$. Common neural language models (e.g., GPT-2/3 [46, 3]) factorize the probability of a sequence into the product of per-token conditionals in left-to-right order, $p_\theta(\mathbf{y}) = \prod_{t=1}^{T} p_\theta(y_t | \mathbf{y}_{<t})$, with each conditional parameterized by a shared neural network, such as transformer [50]. Popular decoding algorithms, ranging from beam search or greedy decoding to sampling methods such as top-$k$ [12] or nucleus [19] sampling, produce text sequences $\mathbf{y}$ using the model $p_\theta$, often conditioned on a prompt $\mathbf{x}$.

**Constrained text generation.** We view text generation as the problem of finding a sequence that satisfies a collection of constraints. For instance, the scenario above amounts to generating a sequence $\mathbf{y} = (y_1, \ldots, y_T)$ subject to a soft constraint that the continuation $\mathbf{y}$ should be fluent and logically coherent with the prompt $\mathbf{x}$. Other constrained generation problems impose additional constraints, such as text infilling [60, 8] where coherence constraints move beyond a left-hand prefix, lexically constrained generation in which hard constraints require the output to contain given tokens, and various forms of semantically-constrained generation in which the output is softly constrained to be

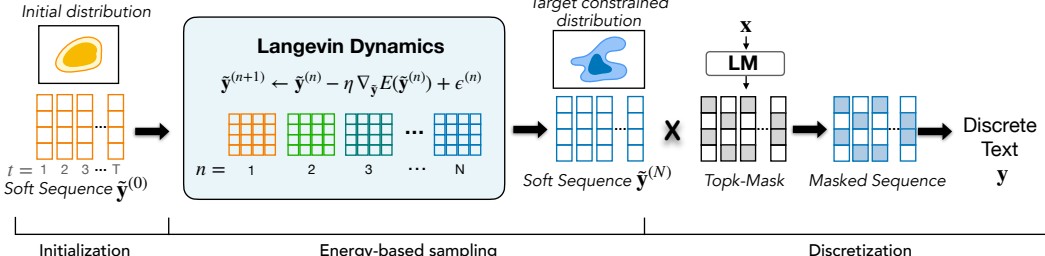

Figure 2: An overview of the COLD decoding procedure. Given an energy function $E(\tilde{\mathbf{y}}) = \sum_i \lambda_i f_i(\tilde{\mathbf{y}})$ with various constraints, the procedure starts with a soft sequence $\tilde{\mathbf{y}}^{(0)}$ as a sample from an initial energy-based distribution, and performs Langevin dynamics iterations using the gradient $\nabla_{\tilde{\mathbf{y}}} E(\tilde{\mathbf{y}})$ (Eq.2). The resulting sequence $\tilde{\mathbf{y}}^{(N)}$ after $N$ iterations is approximately a sample from the desired constrained distribution. We then apply top-k filtering on the soft sequence to produce a discrete text sequence $\mathbf{y}$ (Eq.6).

similar to another sequence. Since common decoding algorithms generate text monotonically, relying on $p_\theta(y_t|\mathbf{y}_{<t})$ for determining the next token, it is challenging to enforce these diverse constraints.

**Energy-based models and Langevin dynamics.** Given an energy function $E(\mathbf{y}) \in \mathbb{R}$, an energy-based model (EBM) is defined as a Boltzmann distribution $p(\mathbf{y}) = \exp\{-E(\mathbf{y})\}/Z$, where $Z = \sum_\mathbf{y} \exp\{-E(\mathbf{y})\}$ is the normalizing factor (The sum is replaced with an integral if $\mathbf{y}$ is continuous). EBMs are flexible, in that one can incorporate arbitrary functions such as constraints into the energy function $E(\mathbf{y})$. Recent work has thus made attempts to *train* text-based EBMs each for specific tasks [21, 41, 7, 23]. As discussed earlier, we instead use the energy-based formulation to develop an *inference* (decoding) procedure that enables off-the-shelf pretrained language models to perform arbitrary constrained generation, without any fine-tuning.

Despite the flexibility, however, sampling from an EBM is particularly challenging, as computing $Z$ is intractable. Common gradient-free Markov chain Monte Carlo (MCMC) methods such as Gibbs sampling [2] can be used, but they are often prohibitively slow [10, 38]. Langevin dynamics [53, 37, 34], a gradient-based MCMC method, offers more efficient sampling by using the gradient of the energy function $\nabla_\mathbf{y} E(\mathbf{y})$, enabling sampling in domains such as image generation [9, 48]. However, since text is discrete, the gradient $\nabla_\mathbf{y} E(\mathbf{y})$ is not well-defined, making it non-trivial to apply Langevin dynamics for sampling text from an EBM. Our approach bridges this gap with continuous relaxation of text, differentiable constraints, and guided discretization, as described below.

## 3 COLD Decoding with Langevin Dynamics

To enable flexible and diverse constrained generation in off-the-shelf language models, we develop *Constrained Decoding with Langevin Dynamics* (COLD), a decoding approach that treats text generation as sampling from an energy-based distribution, allowing for flexibly composing constraints based on the task at hand. COLD decoding generates text by sampling from an EBM defined over a sequence of "soft" tokens using Langevin dynamics, then maps the continuous sample into discrete, fluent text. We provide our formulation of constrained text generation (§3.1), present differentiable constraints that can be composed into energy functions (§3.2) along with our discretization method (§3.3), and discuss practical details of COLD decoding (§3.4). Figure 2 provides an overview.

### 3.1 Energy-based Decoding

Constrained text generation aims to produce text samples $\mathbf{y}$ that satisfy a set of constraints (usually conditioned on an input $\mathbf{x}$ omitted for brevity). We assume each constraint can be captured by a constraint function $f_i(\mathbf{y}) \in \mathbb{R}$, where higher values of $f_i$ mean that the text $\mathbf{y}$ better satisfies the constraint. For example, $f_i$ could measure the likelihood of $\mathbf{y}$ as a fluency constraint (more in §3.2), while a hard constraint $f_i$ amounts to a large negative penalty when $\mathbf{y}$ does not satisfy the constraint.

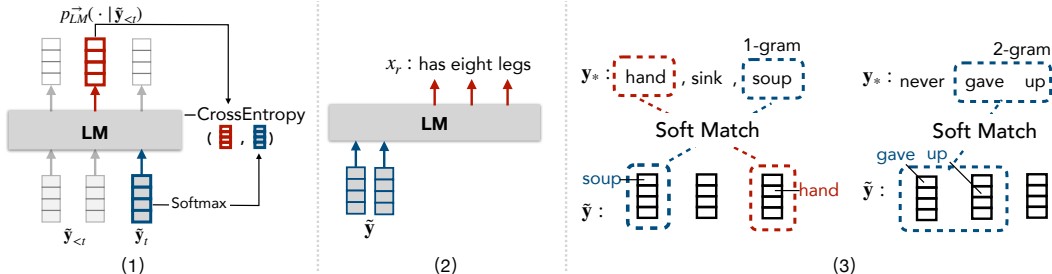

Figure 3: Illustrations of the differentiable constraints introduced in §3.2. **(1)** The soft fluency constraint (Eq.3) to encourage fluency of $\tilde{\mathbf{y}}_t$ based on LM probabilities. **(2)** The future contextualization constraint in Eq.(4) to encourage coherence w.r.t. the future context (has eight legs). **(3)** The $n$-gram similarity constraint in Eq.(5), where the left figure shows the case of $n = 1$ which encourages keywords (e.g., hand) to appear in the generation, and the right figure shows the case of $n > 1$ which is typically used to encourage sequence similarity with a reference text $\mathbf{y}_*$.

The set of constraints induces a distribution over text, written in an energy-based form as:

$$p(\mathbf{y}) = \exp\left\{\sum_i \lambda_i f_i(\mathbf{y})\right\} / Z, \tag{1}$$

where $\lambda_i \geq 0$ is the weight of the $i$th constraint, $Z$ is the normalizing factor. Here $E(\mathbf{y}) := -\sum_i \lambda_i f_i(\mathbf{y})$ is the energy function. This energy-based form is flexible, as one can plug in any constraint functions required for a task of interest. Generating text under the constraints can then be seen as sampling from the energy-based distribution $\mathbf{y} \sim p(\mathbf{y})$. One can also draw multiple samples and pick the best if only one sample is needed, as discussed later (§3.4).

As mentioned above, for efficient sampling from $p(\mathbf{y})$ we want to use Langevin dynamics, which makes use of the gradient $\nabla_{\mathbf{y}} E(\mathbf{y})$. However, in our case $\mathbf{y}$ is a discrete sequence and the gradient $\nabla_{\mathbf{y}} E(\mathbf{y})$ is not well-defined. As a result, we perform Langevin dynamics with an energy defined on a sequence of continuous token vectors, described below.

**Differentiable decoding with Langevin dynamics.** Instead of defining the energy function on discrete tokens, we define the energy function on a sequence of continuous vectors $\tilde{\mathbf{y}} = (\tilde{\mathbf{y}}_1, \ldots, \tilde{\mathbf{y}}_T)$, which we call a soft sequence. Each position in the soft sequence is a vector $\tilde{\mathbf{y}}_t \in \mathbb{R}^V$, where $V$ is the vocabulary size, and each element $\tilde{\mathbf{y}}_t(v) \in \mathbb{R}$ corresponds to the *logit* of word $v$ in the vocabulary. Taking the softmax of $\tilde{\mathbf{y}}_t$ yields a distribution over the vocabulary for position $t$, $\tilde{\mathbf{p}}_t^\tau = \text{softmax}(\tilde{\mathbf{y}}_t/\tau)$. As $\tau \to 0$, $\tilde{\mathbf{p}}_t^\tau$ becomes a one-hot vector, indicating a discrete token.

By specifying an energy $E(\tilde{\mathbf{y}})$ on the soft sequence $\tilde{\mathbf{y}}$, we can use Langevin dynamics to obtain a sample. Specifically, the sampling is done by forming a Markov chain:

$$\tilde{\mathbf{y}}^{(n+1)} \leftarrow \tilde{\mathbf{y}}^{(n)} - \eta \nabla_{\tilde{\mathbf{y}}} E(\tilde{\mathbf{y}}^{(n)}) + \epsilon^{(n)}, \tag{2}$$

where $\eta > 0$ is the step size, and $\epsilon^{(n)} \in \mathcal{N}(0, \sigma)$ is the noise at iteration $n$. As shown in Welling and Teh [53], by adding the right amount of noise and annealing the step size, the procedure will converge to samples from the true distribution. That is, if we let $p^{(n)}$ be the distribution such that $\tilde{\mathbf{y}}^{(n)} \sim p^{(n)}$, then as $n \to \infty$ and $\sigma \to 0$, we have $p^{(n)} \to p(\tilde{\mathbf{y}}) := \exp\{-E(\tilde{\mathbf{y}})\}/Z$. That is, the procedure ends up generating samples from the distribution induced by the energy function.

Next, we describe constraint functions defined on the soft sequence $\tilde{\mathbf{y}}$ that can be plugged in as components of the energy function. Later in §3.3, we describe how to obtain a discrete sequence from a soft sequence sample $\tilde{\mathbf{y}}$.

## 3.2 A Collection of COLD Constraints

COLD provides a flexible framework for plugging in a wide range of constraint functions for a task of interest. We describe constraint functions that are useful in various constrained generation problems, such as those we consider in the experiments (§4). The constraints include language model-based fluency constraints, along with lexical and semantic constraints on the sequence content. More generally, any differentiable function that outputs a goodness score of (soft) text can be used as a constraint function, as long as it reflects the requirements of the target task.

---

**Algorithm 1** Constrained Decoding w/ Langevin Dynamics.

---

**input** Constraints $\{f_i\}$, length $T$, iterations $N$.
**output** Sample sequence $\mathbf{y}$.
$\tilde{\mathbf{y}}_t^{(0)} \leftarrow \texttt{init}()$ for all position $t$ // `init soft-tokens`
    **for** $n \in \{1, \ldots, N\}$ **do**
        $E^{(n)} \leftarrow E(\tilde{\mathbf{y}}^{(n)}; \{f_i\})$ // `compute energy` (§3.2)
        $\tilde{\mathbf{y}}_t^{(n+1)} \leftarrow \tilde{\mathbf{y}}_t^{(n)} - \eta \nabla_{\tilde{\mathbf{y}}_t} E^{(n)} + \epsilon_t^{(n)}$ for all $t$ // `update soft tokens` (Eq.2)
    **end for**
    $y_t = \arg\max_v \texttt{topk-filter}\left(\tilde{\mathbf{y}}_t^{(N)}(v)\right)$ for all $t$ // `discretize` (Eq.6)
**return:** $\mathbf{y} = (y_1, \ldots, y_T)$

---

**Soft fluency constraint.** Fluency is a common requirement for generated text. To promote fluency, we use a constraint which favors soft sequences that receive high probability according to the underlying left-to-right LM $p_{\text{LM}}^{\rightarrow}$ (e.g., GPT2):

$$f_{\text{LM}}^{\rightarrow}(\tilde{\mathbf{y}}) = \sum_{t=1}^{T} \sum_{v \in \mathcal{V}} p_{\text{LM}}^{\rightarrow}(v|\tilde{\mathbf{y}}_{<t}) \log \text{softmax}\left(\tilde{\mathbf{y}}_t(v)\right), \tag{3}$$

where $p_{\text{LM}}^{\rightarrow}(\cdot|\tilde{\mathbf{y}}_{<t})$ means the next-token distribution when providing the neural language model with the preceding soft tokens $\tilde{\mathbf{y}}_{<t}$ (i.e., feeding the weighted average of word embeddings, with the weights being $\text{softmax}(\tilde{\mathbf{y}}_{t'}/\tau)$ for $t' < t$ [20, 45]).

Intuitively, the constraint says that each token distribution in the soft sequence, $\text{softmax}(\tilde{\mathbf{y}}_t)$, must match the "reference" distribution $p_{\text{LM}}^{\rightarrow}(\cdot|\tilde{\mathbf{y}}_{<t})$ predicted by the underlying language model. The match is measured by the (negative) cross-entropy between the two distributions. The constraint thus encourages fluency. In practice, if there is left-side context $\mathbf{x}$ for the generation to condition on, we feed $\mathbf{x}$ to the LM to form the "reference" distribution $p_{\text{LM}}^{\rightarrow}(\cdot|\tilde{\mathbf{y}}_{<t}, \mathbf{x})$. As a result, $\tilde{\mathbf{y}}$ is encouraged to be fluent and coherent with the context $\mathbf{x}$.

We can easily incorporate an additional *reverse* LM constraint, $f_{\text{LM}}^{\leftarrow}$, using a right-to-left LM $p_{\text{LM}}^{\leftarrow}(\cdot|\tilde{\mathbf{y}}_{>t})$, as an additional fluency constraint. Flexibly leveraging multiple models in this way is infeasible with conventional decoding methods such as beam search or nucleus sampling.

**Future-token prediction constraint.** Applications such as text infilling involve future input tokens that remain fixed, but should contribute to updating past positions. For instance, consider updating the second position of `The __ has eight legs`. A sample should be coherent with the tokens $\mathbf{x}_r$ on the right (i.e., `has eight legs`).

To this end, we use a constraint that adjusts soft tokens to maximize the likelihood of input tokens $\mathbf{x}_r$,

$$f_{\text{pred}}(\tilde{\mathbf{y}}; \mathbf{x}_r) = \sum_{k=1}^{K} \log p_{\text{LM}}^{\rightarrow}(x_{r,k}|\tilde{\mathbf{y}}, \mathbf{x}_{r,<k}), \tag{4}$$

where $K$ is the length of $\mathbf{x}_r$. In other words, the constraint adjusts the soft sequence $\tilde{\mathbf{y}}$ such that the underlying LM predicts the future tokens $\mathbf{x}_r$ after seeing $\tilde{\mathbf{y}}$.

**N-gram similarity constraint.** Many constrained generation scenarios pose requirements on the wording and expression of generated text sequences. For instance, lexically constrained generation tasks [18] require certain keywords to be presented in the text samples, while counterfactual reasoning [44] or text editing [15, 31] tasks require the text to retain the essence of a reference sequence.

We formulate these requirements as an $n$-gram similarity constraint which favors sequences that overlap with a reference $\mathbf{y}_*$ at the $n$-gram level,

$$f_{\text{sim}}(\tilde{\mathbf{y}}; \mathbf{y}_*) = \texttt{ngram-match}(\tilde{\mathbf{y}}, \mathbf{y}_*), \tag{5}$$

where $\texttt{ngram-match}(\cdot, \cdot)$ is a recent differentiable $n$-gram matching function [32] which can be seen as a differentiable approximation to the BLEU-$n$ metric [40]. When $n = 1$ and $\mathbf{y}_*$ a sequence of keywords, the constraint in effect enforces $\tilde{\mathbf{y}}$ to assign higher values to the keywords (1-grams). When $n$ is larger and $\tilde{\mathbf{y}}_*$ is a reference sequence, the constraint encourages $\tilde{\mathbf{y}}$ to resemble the reference by assigning high values to tokens making up $n$-grams from $\mathbf{y}_*$.

### 3.3 From Soft to Discrete and Fluent Text

After receiving a soft sequence sample $\tilde{\mathbf{y}}$ from running Langevin dynamics (Eq. 2), we map the soft sequence to a discrete text sequence which we consider as the output of COLD decoding. A simple

| Models | Automatic Eval | | | | Human Eval | | | |
|---|---|---|---|---|---|---|---|---|
| | BLEU$_4$ | ROUGE-L | CIDEr | BERTScore | Grammar | Left-coherence $(\mathbf{x}_l\mathbf{y})$ | Right-coherence $(\mathbf{y}\mathbf{x}_r)$ | Overall-coherence $(\mathbf{x}_l\mathbf{y}\mathbf{x}_r)$ |
| LEFT-ONLY | 0.88 | 16.26 | 3.49 | 38.48 | **4.57** | 3.95 | 2.68 | 2.70 |
| DELOREAN | 1.60 | 19.06 | 7.88 | 41.74 | 4.30 | **4.23** | 2.83 | 2.87 |
| COLD (ours) | **1.79** | **19.50** | **10.68** | **42.67** | 4.44 | 4.00 | **3.06** | **2.96** |

Table 1: Automatic and human evaluation of abductive reasoning (4.1). Our proposed method (COLD decoding) outperforms DELOREAN, a recent decoding algorithm achieving strong results in this task.

method would be selecting the most-likely token at each position $t$, $y_t = \arg\max_v \tilde{\mathbf{y}}_t(v)$. However, the resulting text can suffer from fluency issues even if the soft fluency constraint (Eq. 3) is used, due to competing constraints that sacrifice fluency. To overcome this, we use the underlying LM (e.g., GPT2-XL) as a "guardian" for obtaining the discrete sequence. Specifically, at each position $t$, we first use the LM to produce the top-$k$ most-likely candidate tokens based on its generation distribution conditioning on preceding tokens, which we denote as $\mathcal{V}_t^k$. We then select from the top-$k$ candidates the most likely token based on the soft sample $\tilde{\mathbf{y}}$:

$$y_t = \arg\max_{v \in \mathcal{V}_t^k} \tilde{\mathbf{y}}_t(v). \tag{6}$$

We refer to this method as "top-$k$ filtering". The resulting text tends to be fluent because each token is among the top-$k$ most probable tokens from the LM [12]. In practice, to ease the satisfaction of certain constraints (e.g. $n$-gram similarity), we expand the candidate set $\mathcal{V}_t^k$ to include constraint tokens (e.g., in the tasks of abductive reasoning §4.1 and lexically constrained decoding §4.3).

Figure 2 illustrates the decoding procedure to get one output from COLD decoding. Algorithm 1 summarizes the algorithm. Next, we move to practical considerations of applying COLD.

### 3.4 Implementation of COLD Decoding

**Sample-and-select.** COLD decoding allows for drawing multiple text samples from the distribution induced by the energy function $E(\tilde{\mathbf{y}})$. Depending on task requirements, we could either present the set of samples as output, or select one from the set based on some criteria (e.g., different energy terms) and return a single sequence, as in those tasks considered in the experiments (§4). This "sample-and-select" approach differs from deterministic constrained decoding methods, which optimize only one sequence [e.g., 33, 26], and is used widely in various generation settings [e.g., 28, 11, 4].

**Initialization.** We initialize the soft sequence $\tilde{\mathbf{y}}$ by running greedy decoding with the LM $p_{\text{LM}}$ to obtain output logits. In our preliminary experiments, the initialization strategy had limited influence on the generation results.

**Noise schedule.** Each iteration of Langevin dynamics adds noise $\epsilon^{(n)} \sim \mathcal{N}(0, \sigma^{(n)})$ to the gradient (Eq. 2). We gradually decrease $\sigma^{(n)}$ across iterations, which intuitively transitions the decoding procedure from exploration to optimization. In our experiments, we typically used the schedule which sets/reduces $\sigma$ to $\{1, 0.5, 0.1, 0.05, 0.01\}$ at iterations $\{0, 50, 500, 1000, 1500\}$, respectively.

**Long sequences.** COLD decoding produces a fixed-length sequence $\mathbf{y} = (y_1, \ldots, y_T)$. To produce longer sequences, e.g. in cases where $y_T$ is not the end of a sentence, we use $p_{\text{LM}}$ to produce a continuation of $\mathbf{y}$ using greedy decoding.

## 4 Experiments

We evaluate COLD on three constrained generation tasks. Using COLD for each task amounts to specifying a set of task-specific constraints (instances of those in §3.2). Our focus is enabling constrained generation for settings in which fine-tuning is infeasible, through changing the decoding method. Thus, our experiments (i) use off-the-shelf LMs without fine-tuning, and (ii) compare COLD primarily against alternative decoding methods. As our base LM, we use GPT2-XL [46].

### 4.1 Abductive Reasoning

We study a specific formulation of *abductive reasoning* [43] as a language generation challenge. Specifically, given a beginning sentence $\mathbf{x}_l$ and an ending sentence $\mathbf{x}_r$, the abductive language

generation ($\alpha$NLG) problem [1] consists of generating a bridge sentence $\mathbf{y}$ that fills in between the two sentences and forms a coherent full story (see Figure 1 for example). The task is particularly challenging for conventional monotonic left-to-right LMs (such as GPT-2 and GPT-3) since it requires non-monotonic reasoning that not only conditions on the past context ($\mathbf{x}_l$, on the left), but also the future story ending ($\mathbf{x}_r$, on the right).

### 4.1.1 The COLD Solution

COLD decoding can readily accommodate the abductive reasoning task by simply plugging in appropriate constraints to specify an energy function. Specifically, the generated text needs to be (1) fluent and consistent with the left context $\mathbf{x}_l$, and (2) coherent with the right context $\mathbf{x}_r$. Accordingly, we compose an energy using relevant constraints from §3.2:

$$E(\tilde{\mathbf{y}}) = \lambda_a^{lr} f_{\text{LM}}^{\rightarrow}(\tilde{\mathbf{y}}; \mathbf{x}_l) + \lambda_a^{rl} f_{\text{LM}}^{\leftarrow}(\tilde{\mathbf{y}}; \mathbf{x}_r) + \lambda_b f_{\text{pred}}(\tilde{\mathbf{y}}; \mathbf{x}_r) + \lambda_c f_{\text{sim}}(\tilde{\mathbf{y}}; \text{kw}(\mathbf{x}_r) - \text{kw}(\mathbf{x}_l)). \quad (7)$$

That is, we combine **(a)** a soft fluency constraint (Eq. 3) conditioning on the left sentence $\mathbf{x}_l$ to enforce fluency and consistency with the left context, and a reverse fluency constraint with a right-to-left LM conditioning on $\mathbf{x}_r$ to encourage coherence with the right context; **(b)** a future-token prediction constraint (Eq. 4) that enforces consistency between the generation $\mathbf{y}$ and the story ending $\mathbf{x}_r$; **(c)** a 1-gram similarity constraint (Eq. 5) between the generation $\mathbf{y}$ and keywords (non-stopwords) in $\mathbf{x}_r$ (excluding those in $\mathbf{x}_l$), i.e., $\text{kw}(\mathbf{x}_r) - \text{kw}(\mathbf{x}_l)$, which intuitively promotes a 'smooth transition' between $\mathbf{x}_l$, $\mathbf{y}$, and $\mathbf{x}_r$.

For the energy function in Eq.(7), we select the constraint weights on the dev set. Throughout the experiments, we set the number of Langevin dynamics steps to $N = 2000$, with a step size $\eta = 0.1$ (Eq. 2). We discuss more details of the configurations in the appendix.

**Baselines.** We compare with previous decoding approaches for this task. In particular, we compare with DELOREAN [45] which outperformed a wide range of supervised and unsupervised methods on the abductive reasoning task in Qin et al. [45]. Following Qin et al. [45], we also compare with a LEFT-ONLY method that generates the continuation of $\mathbf{x}_l$ without considering the right-side $\mathbf{x}_r$, i.e., $\mathbf{y} \sim p_{\text{LM}}(\mathbf{y}|\mathbf{x}_l)$.

**Evaluation.** We perform both automatic and human evaluation. We adopt the standard automatic metrics on the task [1] that measure the minimal edit between the generated text and the human-written references on the test set, including BLEU [40], ROUGE [30], CIDEr [51], and BERTScore [58]. For the human evaluation, we follow [45] and let crowdworkers from Amazon Mechanical Turk rate the generations on 200 test examples. Workers were presented a pair of observations ($\mathbf{x}_l$ and $\mathbf{x}_r$) and a generated hypothesis $\mathbf{y}$, and asked to rate the coherence of the hypothesis with respect to the observation $\mathbf{x}_l$ (i.e., $\mathbf{x}_l\mathbf{y}$), the observation $\mathbf{x}_r$ (i.e., $\mathbf{y}\mathbf{x}_r$), and both (i.e., $\mathbf{x}_l\mathbf{y}\mathbf{x}_r$), as well as the grammaticality of the hypothesis $\mathbf{y}$ itself, on a 5-point Likert scale. The average ordinal Krippendorff alpha ($0 \le \alpha \le 1$) [25] is 0.36, indicating a fair inner-annotator agreement.

### 4.1.2 Results

Table 1 shows the evaluation results on the abductive reasoning task. Under automatic evaluation (the left panel), COLD consistently outperforms the previous best unsupervised decoding algorithm DELOREAN, as well as the LEFT-ONLY method, in terms of both the lexical overlap metrics (BLEU, ROUGE and CIDEr) and semantic similarity metric BERTScore. The human evaluation (the right panel) provide more fine-grained insights. **COLD achieves the best overall coherence**, meaning that the generated $\mathbf{y}$ from COLD fits best with both the left-side context $\mathbf{x}_l$ and the right-side context $\mathbf{x}_r$ compared to the other methods. In contrast, DELOREAN excels only in terms of the left-side coherence (with $\mathbf{x}_l$), with inferior right-coherence (with $\mathbf{x}_r$). We speculate this is because of DELOREAN's complex interleaving of forward and backward decoding passes that make it difficult to balance the left- and right-coherence constraints. In terms of grammaticality, unsurprisingly, LEFT-ONLY obtains the best score as it ignores any other constraints (and fails this task with low coherence scores). More importantly, **COLD achieves a high grammaticality score** along with its high coherence, substantially improving over DELOREAN. Example generations in Appendix Table 7 show how COLD can reason with the right-hand context (e.g. 'no heels'), while DELOREAN's generations are contradictory ('red shoes' vs. 'white pair') or equivalent to those from LEFT-ONLY.

| Models | Min-Edit | | Coherence | |
|---|---|---|---|---|
| | Overlap | Human | BERTS. | Human |
| LEFT-ONLY | 50.56 | 1.21 | 73.83 | 2.30 |
| Mix-Match [36] | 85.07 | – | 65.20 | – |
| Mix-Match$_L$ [36] | 84.79 | – | 66.03 | – |
| DELOREAN | 52.90 | 1.81 | 73.66 | 1.92 |
| COLD (ours) | **56.84** | **1.82** | 73.47 | **2.12** |

Table 2: Automatic and human evaluation of counterfactual story rewriting. As a trivial method, LEFT-ONLY is coherent but fails on minimal-edit. COLD is superior to DELOREAN in terms of most metrics, including human evaluation.

| Models | Coverage | | Fluency | |
|---|---|---|---|---|
| | Count | Percent | PPL | Human |
| TSMH | 2.72 | 71.27 | 1545.15 | 1.72 |
| NEUROLOGIC | 3.30 | 91.00 | **28.61** | **2.53** |
| COLD (ours) | **4.24** | **94.50** | 54.98 | 2.07 |

Table 3: Results of lexically constrained decoding (§4.3). For keyword coverage, we report both the average number and average percentage of constraint words present in the generated text. For language fluency, we use perplexity and human judgement.

## 4.2 Counterfactual Story Rewriting

Next, we consider counterfactual story rewriting [44]. Given a story context $\mathbf{x}_l$ with ending $\mathbf{x}_r$, the task is to generate a new story ending $\mathbf{y}$ that is (i) similar to the original ending $\mathbf{x}_r$, yet (ii) consistent with a new story context $\mathbf{x}'_l$ (see Figure 1 for example). The task is challenging as it requires capturing the aspects of future events that are invariant under the new (counterfactual) context, while only making necessary edits for coherence.

### 4.2.1 The COLD Solution

To tackle this task, we use COLD with an energy composed of constraint functions that promote coherence with the new context $\mathbf{x}'_l$, and minimal edits to the original ending $\mathbf{x}_r$:

$$E(\tilde{\mathbf{y}}) = \lambda_a^{lr} f_{\text{LM}}^{\rightarrow}(\tilde{\mathbf{y}}; \mathbf{x}'_l) + \lambda_a^{rl} f_{\text{LM}}^{\leftarrow}(\tilde{\mathbf{y}}) + \lambda_b f_{\text{sim}}(\tilde{\mathbf{y}}; \mathbf{x}_r). \tag{8}$$

These constraints combine: **(a)** a soft fluency constraint (Eq. 3) conditioned on $\mathbf{x}'_l$ to promote coherence between the generation $\mathbf{y}$ and the new (counterfactual) context $\mathbf{x}'_l$; a reverse LM constraint to improve fluency; **(b)** a $n$-gram similarity constraint (Eq. 5, $n = \{2, 3\}$) to encourage generating an ending $\tilde{\mathbf{y}}$ that is close to the original ending $\mathbf{x}_r$. We largely follow the configurations in §4.1 with some exceptions described in the appendix.

**Baselines.** Similar to the setup in §4.1, we compare with DELOREAN [45], a recent state-of-the-art decoding algorithm. As a reference, we also report the performance of a trivial solution, LEFT-ONLY, that generates a continuation of $\mathbf{x}'_l$ without considering the minimal edit constraint with the original ending $\mathbf{x}_r$. Thus the method is expected to generate a coherent ending which however does not necessarily resemble the original ending. Finally, we compare with Mix-and-Match [36], a recent energy-based decoding method with discrete MCMC sampling, using BERT-base and BERT-Large.

**Evaluation.** We use the benchmark dataset TIMETRAVEL [44]. The original data contains three sentences in a story ending. Due to computation constraints, we use the first sentence as the original ending and generate a new single-sentence ending accordingly. Following [44, 45] we conduct both automatic and human evaluation. For automatic evaluation, we measure BERTScore [58], and Minimal Edit which computes the overlap of text edits (insertion, deletion, replacement, etc.) [49] needed to produce the gold ending $\mathbf{y}_*$ and the generated ending $\mathbf{y}$, starting from the original ending $\mathbf{x}_r$. We do not use other common metrics such as BLEU since they were shown to be ineffective [44]. For human evaluation, each crowdworker is presented with the original story $(\mathbf{x}_l, \mathbf{x}_r)$, the counterfactual condition $\mathbf{x}'_l$, and the generated ending $\mathbf{y}$, and the worker is asked to rate (1) the coherence of $\tilde{\mathbf{y}}$ with respect to $\mathbf{x}'_l$ and (2) the extent to which the generated ending $\mathbf{y}$ preserves the details of the original ending $\mathbf{x}_r$ ("minimal edit"), on a 3-point Likert scale for 200 test examples. The average ordinal Krippendorff alpha is 0.52, indicating a moderate inner-annotator agreement. We exclude Mix-and-Match from human evaluation given the significant performance gap in automated evaluation.

### 4.2.2 Results

Table 2 shows the results of automatic and human evaluation in terms of both minimal-edit and coherence. As expected, the reference method LEFT-ONLY that completely ignores the minimal edit constraint can easily generate a new ending that is coherent with the new context $\mathbf{x}'_l$. Compared to the baseline approach DELOREAN, our method COLD achieves overall superior performance, with

substantially improved coherence score and comparable minimal-edit score by human evaluation. Mix-and-Match, based on discrete MCMC sampling, performs poorly. Intuitively, its discrete sampling tends to get stuck in a mode of the target distribution (i.e., the region surrounding the original story ending), and struggles to explore further to find samples of interest. COLD's gradient-based sampling with continuous approximation leads to more efficient and effective exploration and mixing, as evidenced by samples that better meet the task requirements. See Appendix for examples.

## 4.3 Lexically Constrained Decoding

Next, we use COLD for lexically constrained decoding. Given a set of words $\mathcal{W}$, the task aims to generate a coherent sentence that contains these words (Figure 1). The task is challenging as it requires proper planning to coherently include the constraint words.

### 4.3.1 The COLD Solution

We specify an energy function of the following form:

$$E(\tilde{\mathbf{y}}) = \lambda_a^{lr} f_{\text{LM}}^{\rightarrow}(\tilde{\mathbf{y}}) + \lambda_a^{rl} f_{\text{LM}}^{\leftarrow}(\tilde{\mathbf{y}}) + \lambda_b f_{\text{sim}}(\tilde{\mathbf{y}}; \mathcal{W}) + \lambda_c f_{\text{pred}}(\tilde{\mathbf{y}}; c(\mathcal{W})). \tag{9}$$

Specifically, this energy function incorporates: **(a)** a soft fluency constraint (Eq. 3) and a reverse LM fluency constraint as in the previous tasks; **(b)** a 1-gram similarity constraint (Eq. 5) between the generation $\tilde{\mathbf{y}}$ and the given words $\mathcal{W}$; **(c)** a future-token prediction constraint, where we concatenate the constraint words (in an arbitrary order), denoted as $c(\mathcal{W})$, and use it as the right-side content $\mathbf{x}_r$ in Eq. (4). Again we use similar configurations as in §4.1. More details can be found in appendix.

**Baselines.** We compare with a recent state-of-the-art method NEUROLOGIC [33], a beam-search variant specifically designed for lexically constrained generation which outperformed many supervised and unsupervised approaches in Lu et al. [33]. We also report the results of TSMH [57] as another recent baseline which uses Monte-Carlo Tree Search [5].

**Evaluation.** We use the set of constraint words from the COMMONGEN corpus [29], but adopt the *canonical* setting that the generated text must contain the exact constraint words (e.g., write) instead of their variants (e.g., wrote) [18, 47]. Following previous works [18, 47, 57], we report a measure of constraint words coverage as well as language fluency by evaluating the perplexity of the text . We also ask crowdworkers to rate the text fluency on a 3-point Likert scale on 200 test examples. The average ordinal Krippendorff alpha is 0.29, indicating a fair inner-annotator agreement.

### 4.3.2 Results

Table 3 shows the evaluation results for the lexically constrained decoding task. COLD, a *general* constrained decoding method, is comparable to the state-of-the-art method NEUROLOGIC designed specifically for dealing with lexical constraints. In particular, COLD achieves a **higher coverage** of given keywords, at the expense of generating slightly less fluent language. COLD is also substantially better than lexically constrained decoding method TSMH in terms of both coverage and fluency.

## 4.4 Additional Analysis

**Ablation studies.** We ablate two important ingredients of our approach, namely the constraints and the top-$k$ filtering. Due to space limit, we report the results of constraints and defer the results of top-$k$ filtering to the appendix. Table 5 shows the human evaluation results for ablations of the constraints used on the abductive reasoning task (Eq. 7). The $n$-gram similarity constraint $f_{\text{sim}}$ provides the largest contribution to the overall coherence. The reverse LM fluency constraint $f_{\text{LM}}^{\leftarrow}$ also to some extent helps with the right-side coherence by conditioning on the right-side content $\mathbf{x}_r$. Removing the future-token prediction constraint similarly causes inferior scores in terms of right-side and

| Models | Gra-mmar | Left-coher. (x-y) | Right-coher. (y-z) | Overall-coher. (x-y-z) |
|---|---|---|---|---|
| COLD (Full) | 4.17 | 3.96 | **2.88** | **2.83** |
| COLD $-f_{\text{sim}}$ | 4.54 | 3.82 | 2.73 | 2.69 |
| COLD $-f_{\text{LM}}^{\leftarrow}$ | 4.35 | 3.97 | 2.84 | 2.80 |
| COLD $-f_{\text{pred}}$ | **4.61** | **4.07** | 2.75 | 2.77 |

Table 4: Ablation for the effect of different constraints in Eq.(7). We do human evaluation on 125 test examples. The best overall coherence is achieved when all the constraints are present.

overall coherence, as expected. Removing the individual constraints leads to better grammaticality due to less competition among different constraints, at the cost of coherence. Our uniform treatment of all constraints as energy terms makes it straightforward to balance the different constraints by controlling the constraint weights.

**Efficiency of COLD.** We report the average runtime of generating one sample on the Counterfactual Story Rewriting data. The table below shows the results (on an NVIDIA Quadro GV100 GPU, batch size=32). We compare with Mix-and-Match [36], a recent energy-based decoding method with discrete MCMC sampling (Metropolis-Hastings, in particular). COLD, which uses gradient-based sampling, is faster than the gradient-free Mix-and-Match: COLD is 30% faster with base LMs of similar sizes (GPT2-M and BERTLarge), and has roughly the same time cost when using a much larger LM (GPT2-XL).

| Method | Runtime (s) |
|---|---|
| COLD (GPT2-XL, 1.5B) | 33.6 |
| COLD (GPT2-M, 355M) | 22.7 |
| Mix-and-Match (BERTLarge, 340M) | 33.5 |

Table 5: COLD is more efficient than gradient-free Mix-and-Match [36]. The runtime shown is seconds per sample on Counterfactual Story Rewriting.

## 5 Related Work

Previous works proposed beam search variants for lexically constrained decoding [18, 42, 33] which enforce constraints in a discrete space. Recent works consider constraint satisfaction by adjusting vocabulary distributions using an additional discriminator or LM [6, 24, 56]. Differing from those approaches that determine the generation token by token auto-regressively, Qin et al. [45] optimize the whole (soft) token sequence via gradient propagation, which facilitates sequence-level semantic constraints (e.g., right-coherence, minimal-edits). COLD also samples complete sequences, while offering a principled and unified formulation based on energy-based modeling. Kumar et al. [26] extend [17] by imposing constraints with a Lagrangian method and optimizing for a single output with gradient descent. In contrast, our approach based on energy-based sampling (§3.1) allows for generating samples for other utilities (e.g., rank-and-select §3.4, estimating expectations). We also introduce components for more fluent generations such as the novel discretization procedure. Also, on the empirical side, we explore a different class of problems and tackle them in the absence of labeled data. The recent CGMH [35] and TSMH [57], followed by [36, 14], perform constrained decoding with extended Gibbs sampling or Metropolis-Hastings sampling in the discrete text space. Our energy-based formulation with gradient-based Langevin dynamics sampling produces substantially better results than the discrete TSMH (§4.3). Sha [47] uses gradient information to guide generation, which, however, is specifically designed for lexically constrained generation.

Energy-based models (EBMs) have been used for incorporating additional information to train text generation models [7, 23, 41, 21]. In contrast, we focus on the constrained decoding (*inference*) that can be directly applied to pretrained LMs without fine-tuning. Langevin dynamics is widely used on EBMs of modalities with continuous values, like images [48, 9, 59], 3D shapes [55], latent features [39], and audio sequences [22]. To our knowledge, we are the first to apply Langevin dynamics for (constrained) discrete text generation (with a continuous approximation) for efficient sampling.

## 6 Conclusion

We introduce COLD decoding, an energy-based constrained text generation framework that can express various soft/hard constraints through an energy function, and sample using Langevin dynamics. COLD can be applied directly to off-the-shelf LMs without task-specific fine-tuning. We showcase its flexibility and strong performance on three distinct applications of constrained text generation.

## Acknowledgements

This work was funded in part by the Natural Sciences and Engineering Research Council of Canada (NSERC) (funding reference number 401233309), DARPA MCS program through NIWC Pacific (N66001-19-2-4031), the Allen Institute for AI, and Microsoft Research PhD Fellowship. We thank the XLab research group, and our anonymous reviewers for their feedback on this work. We also acknowledge the Beaker team (`https://beaker.org`) for their support with experiments.

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
