# OpenReview forum: "COLD Decoding: Energy-based Constrained Text Generation with Langevin Dynamics"
_NeurIPS.cc/2022/Conference — NeurIPS 2022 Accept_

### Official Review · Reviewer_zcYZ · 2022-07-11

**Rating:** 8
**Confidence:** 3
**Soundness:** 4 excellent
**Presentation:** 4 excellent
**Contribution:** 4 excellent

**Summary:**

This paper proposes a generative model for constrained text generation. It incorporates hard and soft constraints in text generation and experiments on three tasks, lexically constrained generation, abducting reasoning, and counterfactual reasoning. The approach formulates generation as sampling from an energy-based model. Those constraints are applied as the weighted components for the score function. They introduced Langevin dynamics for efficient gradient-based sampling.


**Questions:**

I wonder how fast is the proposed algorithm?

I am not 100% sure about the "sample-and-select". How many samples do you draw for one example? Any more detail about this sampling procedure?

**Limitations:**

I am mostly worried about the limited scope of comparison.

**Strengths And Weaknesses:**

Strength
The paper presents a novel approach to conditional/constrained text generation. Langevin dynamics has been explored in machine learning and computer vision but it’s novel to NLP and text generation. The approach can model different kind of constraints, hard or soft, in a unified approach. It is grounded with solid foundation and potential use cases.
The paper is well written and presented. It’s a fairly complicated topic, but examples and figures made the paper easier to follow.
The approach is validated on three NLG tasks, and the approach beats prior work like DELOREAN. Weakness
Although the experiments cover three tasks, the scope of comparison is limited. It mostly compares with one baseline, Left-Only, and one model DeLorean. The author of this paper, DeLorean and NeuroLogic are essentially from the same research group, so I am slightly concerned about the credibility of the experimental results. It’s totally possible to compare with prior work using these 3 datasets. The authors can also find some other tasks like sentiment transfer or attribute grounded generation. The FUDGE paper and the Mix and Match paper are good references to look at.


Some missing reference
The mix and match paper is directly relevant; the FUDGE paper (and many other discriminator based constrained NLG papers) are also worth mentioning and comparing with.
FUDGE: Controlled Text Generation With Future Discriminators
https://aclanthology.org/2021.naacl-main.276.pdf
Mix and Match: Learning-free Controllable Text Generation using Energy Language Models
https://aclanthology.org/2022.acl-long.31.pdf

---

> ### Author Response · Authors · 2022-08-02
> **Author Response**
>
> Thank you for your encouraging comments and suggestions.
>
> **Missing references and comparison**\
> Thanks for the suggestions. We’ve indeed already cited and discussed Mix-and-Match and related gradient-free MCMC based papers in our Related Work section in lines 359-362 (also notice that the Mix-and-Match paper was made public on March 24, only slightly over one month before the NeurIPS submission). We’ve also discussed related discriminator based constrained NLG papers such as PPLM [6]. We’ll add the reference to FUDGE and discuss – thanks for the pointer!
>
> As suggested by the reviewer, in the table below, we add comparison with Mix-and-Match on the Counterfactual Story Rewriting task. Recall that the task aims at *minimally editing* a given story ending to re-gain the *coherence* with the counterfactual story context (Section 4.2). We found that Mix-and-Match, based on the discrete MCMC (Metropolis-Hastings) sampling, has difficulty in mixing well to generate the desired samples on the task.
>
> Specifically, we used both BERT-base and BERT-large models provided in the Mix-and-Match official code. Only imposing the coherence constraint (i.e., without any minimal-edit constraints) gives the results in the table, where we can see the method fails to edit the original story ending sufficiently to achieve coherence. Adding the minimal-edit constraints would only make the situation even worse; and increasing the model size does not alleviate the issue much as shown in the table . This is because the discrete MCMC sampling tends to be stuck in a mode (i.e., the region surrounding the original story ending) of the target distribution and struggle to explore further to find the samples of interest. In comparison, our gradient-based sampling with continuous approximation leads to more efficient and effective exploration/mixing, and hence better samples that meet the task requirements. We’ll add more results and discussion in the paper.
>
> | Method  | Min-edit  |  Coherence |
> |---|---|---|
> | COLD  | 56.84  | 73.47  |
> | Mix-and-Match (base) |  85.07  | 65.20  |
> | Mix-and-Match (large) |  84.79  | 66.03  |
>
> &nbsp;
>
> **How fast is the proposed algorithm**\
> We report the average runtime of generating one sample on the Counterfactual Story Rewriting data, and compare with Mix-and-Match. The table below shows the results (on an NVIDIA Quadro GV100 GPU, batch size=32). We can see that COLD with the gradient-based sampling, is faster than the gradient-free alternative Mix-and-Match. COLD is 30% faster with the base LMs of similar sizes, and has roughly the same time cost when using a larger LM (while achieving much better performance as above).
>
> | Method  | Runtime (s) per sample |
> |---|---|
> | COLD (GPT2-XL, 1.5B)  | 33.6  |
> | COLD (GPT2-M, 355M)  | 22.7  |
> | Mix-and-Match (BERTLarge, 340M)  | 33.5  |
>
> &nbsp;
>
> **DeLorean and NeuralLogic as baselines**\
> We chose DeLorean and NeuralLogic as the primary baselines on the respective tasks, because they have shown to outperform a wide range of other methods in their studies. We’ll add the additional results as above in the paper. Thanks for the suggestion.
>
> &nbsp;
>
> **sample-and-select** \
> We drew 16 to 64 samples for one example in different tasks. Sampling is straightforward by the nature of Langevin dynamics (LD), as we simply run the LD chain multiple times, which in our implementation is done in parallel in a batch way and is thus efficient. We’ll add these details in the revision.

---

> > ### Comment · Reviewer_zcYZ · 2022-08-03
> > **reviewer update after author response**
> >
> > Hi authors, thanks for your detailed response with extra experiments and clarification. I am very happy about the response and I believe this is a strong submission. I hope the authors can update the results accordingly in the final version of the paper. I am open to discussion.
> > Update on rating: 7 -> 8.

---

> > > ### Author Response · Authors · 2022-08-07
> > > **Thank you**
> > >
> > > Thank you for raising the score! We'll incorporate the new results & comparison in the revised version!

---

### Official Review · Reviewer_FCnJ · 2022-07-11

**Rating:** 7
**Confidence:** 4
**Soundness:** 3 good
**Presentation:** 3 good
**Contribution:** 3 good

**Summary:**

Many applications of text generation require incorporating different constraints (hard or soft constraints) to control the semantics or style of generated text. The authors propose a decoding framework named Energy-based Constrained Decoding with Langevin Dynamics, which can achieve hard and soft constraints in a unified framework and does not need any task-specific fine-tuning. Experimental results on three tasks of constrained text generation show the effectiveness of proposed framework.

**Questions:**

What is the detail of obtaining a right-to-left LM?

**Strengths And Weaknesses:**

Strengths
- The authors propose a decoding framework to unify hard and soft constrained generation. The framework treats constrained generation as an energy function and is flexible to add any constraints through arbitrary constraint functions.
- The authors propose a new sampling method to achieve efficient search for a single optimal solution.
- Experiments on three constrained generation tasks show that the proposed framework achieves strong performance on both automatic and human evaluation.

Weaknesses
- Consider that 2,000 iterations are required, the author should give the time consumed by the method to generate the text.
- Considering that GPT2 itself may generate many candidates through top-k sampling, the authors should add a baseline, in which sample-and-select is performed on generated text of GPT2 by top-k sampling.

---

> ### Author Response · Authors · 2022-08-02
> **Author Response**
>
> Thank you for your positive feedback and suggestions.
>
> **Runtime of generation**\
> We report the average runtime of generating one sample on the Counterfactual Story Rewriting data. The table below shows the results (on an NVIDIA Quadro GV100 GPU, batch size=32). We compare with Mix-and-Match [[Mireshghallah, et al., ACL 2022]](https://arxiv.org/abs/2203.13299), a latest energy-based decoding method with discrete MCMC sampling (Metropolis-Hastings, in particular). We can see that our COLD with the gradient-based sampling, is faster than the gradient-free Mix-and-Match. COLD is 30% faster with the base LMs of similar sizes (GPT2-M and BERTLarge), and has roughly the same time cost when using a much larger LM, GPT2-XL (while achieving much better performance as shown in our response to Reviewer zcYZ).
>
> | Method  | Runtime (s) per sample |
> |---|---|
> | COLD (GPT2-XL, 1.5B)  | 33.6  |
> | COLD (GPT2-M, 355M)  | 22.7  |
> | Mix-and-Match (BERTLarge, 340M)  | 33.5  |
>
> &nbsp;
>
> **top-k sampling baseline** \
> Thanks for the suggestion. We note that the top-k sample-and-select baseline was already compared by DeLorean [45] (https://arxiv.org/pdf/2010.05906.pdf), where DeLorean is shown to outperform the top-k baseline consistently on different tasks (e.g., Table.1 in [45]). Hence in our paper, we compare directly with the DeLorean as a much stronger baseline. We’ll make this clear and add the top-k sample-and-select results in the revision.
>
> &nbsp;
>
> **Details of obtaining a right-to-left LM**\
> Thanks for the question. The right-to-left LM is publicly released by the paper [West et al., ACL 2021] (https://arxiv.org/abs/2010.08566). Specifically, the LM was trained following GPT-2 using the OpenWebText training corpus (see section 2.4 in [West et al., ACL 2021]). We’ll add the details in the revision.

---

### Official Review · Reviewer_5VZW · 2022-07-11

**Rating:** 8
**Confidence:** 3
**Soundness:** 4 excellent
**Presentation:** 3 good
**Contribution:** 4 excellent

**Summary:**

This work proposes a constrained generation decoding plug-in framework that does not require additional fine-tuning and can be applied to any left-to-right LM. The authors used sampling from the EBM model to incorporate constraints, and an energy function is specified by defining relevant constraint functions. One of the main novelties of the paper is that it suggests Langevin dynamics for gradient-based efficient sampling. That allows applying the framework to various constrained tasks. Experiments on three different tasks show that the proposed method outperforms previous works on constrained generation.

**Questions:**

Q1: Did I correctly understand that the constraint function must be handcrafted, and it has a significant impact on the performance? Do you have any suggestions on how to propose a new constrain function?

Q2:  Efficiency of the sampling is claimed in the paper. However, there are no insights on the decoding speed/latency and comparison with other methods, if possible

Q3: Is it possible to incorporate several constraints?


**Limitations:**

-

**Strengths And Weaknesses:**

Strengths:
- Unified decoding framework for constrained generation
- The method is well-motivated and described carefully
- Shows solid improvements upon over methods

Weaknesses:
- The analysis of the results lacks some interesting aspects (see questions)

---

> ### Author Response · Authors · 2022-08-02
> **Author Response**
>
> Thank you for recognizing the novelties and advantages of our method.
>
> **Q1: constraint function** \
> Our approach provides a flexible framework for users to plug in a wide range of constraint functions for a wide range of tasks. To use our method, a user would design or pick the constraint functions for their task. Any differentiable function that outputs a goodness score of (soft) text can be used as a constraint function, as long as it reflects the requirements of the target task (e.g., we designed the *n-gram similarity constraint* for the “minimal-edit” requirement of the “counterfactual story rewriting” task). For example, many task evaluation metrics (e.g., BERTScore) can be used as constraint functions. In the paper, we’ve proposed a set of constraint functions that are commonly useful in many tasks (e.g., the *soft fluency constraint* for potentially all generation tasks, and the above-mentioned *n-gram similarity constraint* for “minimal-edit” in counterfactual generation or text style transfer tasks).
>
> &nbsp;
>
> **Q2: Efficiency of the sampling** \
> We report the average runtime of generating one sample on the Counterfactual Story Rewriting data. The table below shows the results (on an NVIDIA Quadro GV100 GPU, batch size=32). We compare with Mix-and-Match [[Mireshghallah, et al., ACL 2022]](https://arxiv.org/abs/2203.13299), a latest energy-based decoding method with discrete MCMC sampling (Metropolis-Hastings, in particular). We can see that our COLD with the gradient-based sampling, is faster than the gradient-free Mix-and-Match. COLD is 30% faster with the base LMs of similar sizes (GPT2-M and BERTLarge), and has roughly the same time cost when using a much larger LM, GPT2-XL (while achieving much better performance as shown in our response to Reviewer zcYZ).
>
> | Method  | Runtime (s) per sample |
> |---|---|
> | COLD (GPT2-XL, 1.5B)  | 33.6  |
> | COLD (GPT2-M, 355M)  | 22.7  |
> | Mix-and-Match (BERTLarge, 340M)  | 33.5  |
>
> &nbsp;
>
> **Q3: incorporating several constraints** \
> Our method is designed to allow incorporating an arbitrary number of constraints. Users can plug in and combine them for their tasks. For example, in each of the three tasks in our experiments, we incorporated several constraints (e.g., the four constraints in Eq.7 for abductive reasoning).

---

> > ### Comment · Reviewer_5VZW · 2022-08-07
> > **Reviewer comment after author response**
> >
> > Dear authors,
> >
> > Thank you for clarification and providing additional results. I remain positive on my previous assessment.
> >
> > Best,
> > Reviewer 5VZW

---

### Official Review · Reviewer_EAUi · 2022-07-18

**Rating:** 6
**Confidence:** 3
**Soundness:** 3 good
**Presentation:** 2 fair
**Contribution:** 3 good

**Summary:**

The paper proposes a new framework for energy-based constrained text generation. The proposed method implements text generation as gradient based sampling from an energy-based distribution, derived as a linear composition of the distribution defining each constraint. The method description is clear. Paper is well-written and usually easy to follow, although please see comments for suggested revisions.

Method is evaluated in three different tasks requiring constrained generation and obtains promising results in comparison to previous approaches. Comparison to related work could be more extensive and through. Overall, an interesting idea that should inspire other research directions in the related field.

**Questions:**

Q1. Evaluation of results and comparison to baselines is a bit superficial. Did the authors do any analysis to see the differences in characteristics of the outputs generated by different models?

Q2. Table 2: why are the BLEU results so low?

**Limitations:**

adequate

**Strengths And Weaknesses:**

Comments:

- line 17-24: discussion and overview of the problem is introduced too generally, and the examples do not directly connect to the figure. A revision can make more clear how the method is directly applicable to each task.

- line 25- 27: the discussion on supervised learning is too generic. The proposed method does not necessarily solve the annotation scarcity. There also many applications where it is feasible to adapt / finetune models with domain specific annotations

- line 100: token "next" used unnecessary and confusing.

---

> ### Author Response · Authors · 2022-08-02
> **Author Response**
>
> Thank you for your encouraging comments.
>
> We’ll make the discussion on the problem overview and supervised learning clearer. Specifically,
>
>   - **Line 12-24**: The three tasks in Figure 1 are examples of the problems mentioned in Line 17-24. For example, *Lexically Constrained Generation* in the figure is an example of the “keyword-guided generation” in the discussion; the “abductively reasoning about what happened in the middle of a story” in the text refers to the *Abductive Reasoning* task in the figure; while “revising an input based on a new counterfactual condition” refers to the *Counterfactual Reasoning* task. Thus the discussion in Line 17-24 connects to the illustrations in the figure. We’ll make this clearer.
>
>   - **Line 25-27**: We meant to discuss specifically the supervised learning for the combinatorially many *constrained text generation* tasks where annotated data is often unavailable. Our approach does not need to train specialized models for the specific tasks, and alleviates the data scarcity problem to some extent. We’ll revise the discussion to make it more specific and avoid confusion.
>
>   - **Line 100**: We’ll remove the token “next” to make it clearer.
>
> &nbsp;
>
> **Analysis of the outputs** \
> We performed human evaluation on the different characteristics of the model outputs. For example, in the abductive reasoning task (Section 4.1), we evaluate the outputs in terms of the grammaticality, and coherence w.r.t left context, right context, and the overall context, respectively. Those aspect-based evaluations offer insights into the strengths and weaknesses of the different models (e.g., our approach tends to generate more grammatical text than DeLorean thanks to the better sampling approach). Besides, in the supplementary materials (Appendix E), we offer example outputs from different models on various tasks, to give a qualitative sense of the results. We’ll add more examples and analysis in the revision.
>
> &nbsp;
>
> **BLEU results** \
> We clarify that Table 2 does not include BLEU results. We thought the reviewer was referring to Table 1 which reports the BLEU-4 scores. The scale of the BLEU-4 scores is low, primarily because the task (abductive reasoning) is relatively open-ended and can have different plausible generations to be filled in the context. We notice that, as reported in previous work [45], the human-written text on this task has a BLEU-4 score of 8.25, which is also low (as compared to the typical scale of BLEU-4 on other tasks such as machine translation). Hence the scale of the BLEU score is primarily due to the nature of the task. Our approach improves over the baselines in terms of BLEU-4 and other diverse metrics.

---

### Author Response · Authors · 2022-08-02
**General Response**

We thank the reviewers for their insightful and encouraging comments. The reviewers appreciated the novelty of our approach (“*it’s novel to NLP and text generation and grounded with solid foundation and potential use cases*”), the empirical gains on three tasks (“*shows solid improvements upon over methods*”), and the potential inspiration for additional tasks (“*an interesting idea that should inspire other research directions*”).

We’d like to highlight that among other contributions of the paper, this is the first work that unifies constrained generation as specifying constraints through an energy function, and performs reasoning on constraints through Langevin dynamics sampling. Also, as pointed out by R2 and R3, our method is flexible and can be plugged in with any off-the-shelf language model without the need of task specific fine-tuning.

Below we address the raised concerns, and we will make corresponding modifications in the revised paper.

---

### Meta-Review · Area_Chair_gwEm · 2022-08-25

**Recommendation:** Accept
**Confidence:** Certain

**Metareview:**

This paper proposes a framework for controlled or constrained text generation where the constraints are encoded with energy-based models (EBMs). Generation proceeds in two steps: first, the discrete words are relaxed into continuous vectors of logits (scores), which enables the use of gradient methods and Langevin dynamics to obtain a sample. Then, to obtain actual words (a discrete output) a LM is used for top-k filtering and the word with the largest score is chosen. The paper demonstrates the applicability of the proposed framework in three different generation tasks, controlled generation, abductive, and counterfactual generation.

All reviewers agree (and I agree with them) that this is a solid paper which brings Langevin dynamics (a well-known technique mainly used with continuous outputs, e.g. in vision tasks) to text generation. Although the use of this technique with EBMs is certainly not novel, making it work for text is non-trivial. The reviewers suggest several improvements: reporting the runtimes (the proposed method requires many gradient iterations which brings a considerable slow down), comparing against more baselines more explicit, and adding references and discussion related for recent work such as FUDGE and mix-and-match. The author response was satisfactory and promised to add these details to the final version.

I strongly encourage the authors to incorporate the reviewer's suggestions in their final version.



**Award:**

No

---

### Decision · Program_Chairs · 2022-09-14

Accept